# Long-Term Prevalence of Sensory Chemotherapy-Induced Peripheral Neuropathy for 5 Years after Adjuvant FOLFOX Chemotherapy to Treat Colorectal Cancer: A Multicenter Cross-Sectional Study

**DOI:** 10.3390/jcm9082400

**Published:** 2020-07-27

**Authors:** Marie Selvy, Bruno Pereira, Nicolas Kerckhove, Coralie Gonneau, Gabrielle Feydel, Caroline Pétorin, Agnès Vimal-Baguet, Sergey Melnikov, Sharif Kullab, Mohamed Hebbar, Olivier Bouché, Florian Slimano, Vincent Bourgeois, Valérie Lebrun-Ly, Frédéric Thuillier, Thibault Mazard, David Tavan, Kheir Eddine Benmammar, Brigitte Monange, Mohamed Ramdani, Denis Péré-Vergé, Floriane Huet-Penz, Ahmed Bedjaoui, Florent Genty, Cécile Leyronnas, Jérôme Busserolles, Sophie Trevis, Vincent Pinon, Denis Pezet, David Balayssac

**Affiliations:** 1INSERM U1107 NEURO-DOL, Université Clermont Auvergne, F-63000 Clermont-Ferrand, France; mselvy@chu-clermontferrand.fr (M.S.); nkerckhove@chu-clermontferrand.fr (N.K.); coralie.gonneau@laposte.net (C.G.); jerome.busserolles@uca.fr (J.B.); 2Service de Chirurgie digestive, CHU Clermont-Ferrand, F-63000 Clermont-Ferrand, France; cpetorin@chu-clermontferrand.fr (C.P.); avimal-baguet@chu-clermontferrand.fr (A.V.-B.); dpezet@chu-clermontferrand.fr (D.P.); 3Délégation à la Recherche Clinique et à l’Innovation, CHU Clermont-Ferrand, F-63000 Clermont-Ferrand, France; bpereira@chu-clermontferrand.fr (B.P.); gabrielle.feydel@etu.uca.fr (G.F.); 4Service de pharmacologie médicale, CHU Clermont-Ferrand, F-63000 Clermont-Ferrand, France; 5Centre Hospitalier de Saint-Flour, Service Chirurgie générale et viscérale, F-15100 Saint-Flour, France; smelnikov@ch-stflour.fr; 6Centre Hospitalier de Moulins Yzeure, Service Oncologie, F-03006 Moulins, France; sharif_kullab@hotmail.com; 7CHRU Lille, Service Oncologie, F-59000 Lille, France; mohamed.hebbar@chru-lille.fr; 8CHU Reims, Service Oncologie digestive, Université de Reims Champagne-Ardenne, F-51100 Reims, France; obouche@chu-reims.fr; 9CHU Reims, Service Pharmacie, BioSpecT, EA n 7506, SFR CAP-Santé, Université de Reims Champagne-Ardenne, F-51100 Reims, France; fslimano@chu-reims.fr; 10Centre Hospitalier de Boulogne sur Mer, Service Oncologie digestive, F-62321 Boulogne-Sur-Mer, France; v.bourgeois@ch-boulogne.fr; 11CHU Limoges, Service Oncologie, F-87042 Limoges, France; valerie.ly@chu-limoges.fr (V.L.-L.); frederic.thuillier@chu-limoges.fr (F.T.); 12IRCM, Inserm, Univ Montpellier, ICM, F-34298 Montpellier, France; thibault.mazard@icm.unicancer.fr; 13Infirmerie protestante de Lyon, Service Gastro-entérologie, F-69300 Caluire et Cuire, France; davidtavan@gmail.com; 14Centre Hospitalier Emile Roux, Service Oncologie, F-43000 Le Puy-en-Velay, France; kheireddine.benmammar@ch-lepuy.fr (K.E.B.); brigitte.monange@ch-lepuy.fr (B.M.); 15Centre Hospitalier de Béziers, Service Gastro-entérologie, F-34500 Béziers, France; mohamed.ramdani@ch-beziers.fr; 16Centre Hospitalier Saint-Joseph Saint-Luc, Service Hépato-gastro-entérologie, F-69007 Lyon, France; dpereverge@ch-stjoseph-stluc-lyon.fr; 17Centre Hospitalier Alpes Leman, Service Gastro entérologie, F-74130 Contamine sur Arve, France; fhuet-penz@ch-alpes-leman.fr; 18Centre hospitalier Intercommunal Les Hôpitaux du Léman, Service Gastro-entérologie, F-74203 Thonon les bains, France; a-bedjaoui@ch-hopitauxduleman.fr; 19Centre Hospitalier de Vichy, Service Chirurgie digestive et viscérale, F-03200 Vichy, France; florent.genty@ch-vichy.fr; 20Groupe Hospitalier Mutualiste de Grenoble, Service Oncologie, F-38000 Grenoble, France; c.leyronnas@ghm-grenoble.fr; 21CHU Clermont-Ferrand, Service Pharmacie, Clermont-Ferrand, F-63000 Clermont-Ferrand, France; strevis@chu-clermontferrand.fr (S.T.); vpinon@chu-clermontferrand.fr (V.P.); 22INSERM U1071, M2iSH, Université Clermont Auvergne, USC-INRA 2018, F-63000 Clermont-Ferrand, France

**Keywords:** peripheral neuropathy, oxaliplatin, colorectal cancer, health-related quality of life, cancer survivors

## Abstract

(1) Background: Oxaliplatin is among the most neurotoxic anticancer drugs. Little data are available on the long-term prevalence and consequences of chemotherapy-induced peripheral neuropathy (CIPN), even though the third largest population of cancer survivors is made up of survivors of colorectal cancer. (2) Methods: A multicenter, cross-sectional study was conducted in 16 French centers to assess the prevalence of CIPN, as well as its consequences (neuropathic pain, anxiety, depression, and quality of life) in cancer survivors during the 5 years after the end of adjuvant oxaliplatin chemotherapy. (3) Results: Out of 406 patients, the prevalence of CIPN was 31.3% (95% confidence interval: 26.8–36.0). Little improvement in CIPN was found over the 5 years, and 36.5% of patients with CIPN also had neuropathic pain. CIPN was associated with anxiety, depression, and deterioration of quality of life. None of the patients with CIPN were treated with duloxetine (recommendation from American Society of Clinical Oncology), and only 3.2%, 1.6%, and 1.6% were treated with pregabalin, gabapentin, and amitriptyline, respectively. (4) Conclusions: Five years after the end of chemotherapy, a quarter of patients suffered from CIPN. The present study showed marked psychological distress and uncovered a failure in management in these patients.

## 1. Introduction

Oxaliplatin is a pivotal drug in the management of colorectal cancer. It is combined with 5-fluorouracil and folinic acid in FOLFOX protocol (400 mg/m^2^ intravenous (i.v.) 5-FU bolus, 200 mg/m^2^ i.v. folinic acid, and 85 mg/m^2^ oxaliplatin, followed by a 22-h infusion of 600 mg/m^2^ i.v. 5-FU). After surgery, the FOLFOX protocol is the standard treatment for stage II and III colorectal cancer [1].

Oxaliplatin-induced peripheral neuropathy is a disabling adverse drug reaction that interferes with patients’ quality of life [2]. This chemotherapy-induced peripheral neuropathy (CIPN) manifests itself in a typical stocking-glove pattern, with symptoms of paresthesia and dysesthesia, such as tingling, neuropathic pain, and numbness [2]. In addition, oxaliplatin causes acute cold hypersensitivity during chemotherapy, which is a marker of its neurotoxicity. This cold hypersensitivity decreases after the end of the chemotherapy [3]. Several risk factors have been identified such as cumulative doses >850 mg/m^2^ [4,5,6], pre-treatment anemia, hypoalbuminemia and hypomagnesaemia, alcohol consumption [7], and genetic polymorphisms (voltage-gated sodium channel and cyclin H) [8,9,10]. No treatment can prevent CIPN, and only duloxetine seems to relieve pain symptoms [11]. Consequently, oncologists are frequently forced to decrease or discontinue oxaliplatin doses [12], which may have a negative impact on disease control and progression-free survival [13].

Oxaliplatin is probably the most neurotoxic anticancer drug; in one study, more than 90% of patients receiving the drug developed acute neuropathy, and 30%–50% of patients developed chronic neuropathy during treatment [2]. However, the severity and duration of these symptoms have varied among studies [4]. Another study reported that CIPN assessment based on clinician-reported outcome underestimated the prevalence and severity of the condition [14]. Although symptoms decrease with time, long-term clinical studies have demonstrated that CIPN may persist for more than 12 months and that they may be more common and severe than expected [4,15], raising concerns about the reversibility of the condition [16], in the third largest population of cancer survivors (i.e., colorectal cancer) [17].

The objective of the present study was to assess the prevalence and severity of CIPN after the end of adjuvant oxaliplatin chemotherapy to treat colorectal cancer. In addition, neuropathic pain, anxiety, depression, health-related quality of life (HRQoL), and use of pain medications were assessed.

## 2. Experimental Section

### 2.1. Study Design

The present multicenter, cross-sectional study aimed to assess the prevalence and severity of CIPN in survivors of colorectal cancer for 5 years after the end of oxaliplatin-based chemotherapy, based on self-administered questionnaires. As secondary objectives, the following were also assessed: prevalence of neuropathic pain, prevalence of anxiety and depression, HRQoL, and use of pain medications. Patients were assessed once, and no longitudinal assessment was performed.

The study conformed to the Strengthening the Reporting of Observational Studies in Epidemiology (STROBE) guidelines [18], and the protocol was registered on ClinicalTrials.gov (NCT02970526). The study was approved by a local ethics committee (Comité de Protection des Personnes sud-est 6, IRB: 00008526, No. 2016/CE16, 26/02/2016) and was carried out anonymously. The study was approved by the Advisory Committee on the Treatment of Research Information (No. 15.645, 13/05/2015). Consent was obtained from all participants by telephone.

### 2.2. Setting

The study was coordinated by the University Hospital of Clermont-Ferrand (CHU Clermont-Ferrand, France). Patients were recruited from 16 French centers (University Hospitals: CHU Clermont-Ferrand, CHU Limoges, CHU Reims, CHRU Lille, and Institut du Cancer Montpellier; General Hospitals: CH Saint-Flour, CH Moulins, CH Boulogne-sur-Mer, CH Béziers, CH Puy en Velay, Infirmerie Protestante de Lyon, CH Saint-Joseph Saint Luc Lyon, CH Alpes Leman, CHI Les Hôpitaux du Léman, CH Vichy, and GHM Grenoble). Patients were recruited from 21 June 2016 until 29 August 2019.

### 2.3. Participants

The inclusion criteria were as follows: (1) treatment with adjuvant oxaliplatin-based chemotherapy (FOLFOX-4) for colorectal cancer, (2) ≤5 years between when chemotherapy was discontinued and the survey was completed, and (3) no cancer relapse during these 5 years (cancer survivor). The exclusion criteria were age < 18 years, and patients with neurological diseases (stroke, Parkinson’s disease, Alzheimer’s disease).

Patients were identified from the database of the chemotherapy prescription software of each participating center. Thereafter, according to the inclusion/exclusion criteria, each center phoned their patients to inquire whether they would participate in the study. After patient acceptance, a paper questionnaire and a stamped envelope for the response were sent to the patient. Patients returned their questionnaires to the coordinating center, where their responses were recorded and analyzed.

### 2.4. Variables

The primary endpoint was the sensory score of the EORTC QLQ-CIPN20, which rates CIPN severity from 0 (least) to 100 (worst) during the last week [19] (for scoring see: https://www.eortc.org/app/uploads/sites/2/2018/02/SCmanual.pdf). Sensory CIPN was defined as a sensory QLQ-CIPN20 score of ≥30/100 in the present study, based on a work by Alberti et al. (2014) [20].

With regards to the secondary endpoints, ongoing neuropathic pain was defined as a visual analogue scale (VAS) score ≥40/100 and a DN4 (French abbreviation: Douleur Neuropathique 4, for neuropathic pain 4) interview questionnaire score ≥3/7 [21]. The proportions of patients with ongoing thermal hypersensitivity to either cold or heat was recorded (“do you currently fear contact with cold/hot objects or the ingestion of cold/hot drinks?”), and the hypersensitivity was assessed using a VAS (0 = no hypersensitivity, 100 = maximum imaginable hypersensitivity). Anxiety and depression were assessed using the Hospital Anxiety and Depression Scale (HADS) questionnaire at the time of the answer (normal: ≤7/21, borderline or suggestive of anxiety/depression: 8–10/21, indicative of anxiety/depression: ≥11/21) [22]. The patients’ HRQoL at the time of the answer was assessed using the EORTC QLQ-C30 [23]. Ongoing pain medications in the past month were recorded, as were the patients’ oncological treatment characteristics, including cumulative dose (mg/m^2^), dose intensity (mg/m^2^/week), number of oxaliplatin cycles, mean percentage of cycles with a dose reduction (≥10%), and dates of the first and last oxaliplatin cycles. Socio-demographic characteristics were recorded, including sex, age, daily cigarette use, occasional alcohol use, and hazardous alcohol use (males: ≥21 alcohol units/week and females: ≥14 alcohol units/week), at the time of the answer. Height, weight, body mass index (BMI), and body surface area were recorded from chemotherapy prescription software (data during chemotherapy treatment).

### 2.5. Data Sources and Measurements 

Data assessing CIPN, neuropathic pain, anxiety, depression, HRQoL, and ongoing pain medications were obtained from the completed questionnaire. Oncological data and patient characteristics were obtained from the chemotherapy prescription software of each center. All the data were recorded and managed using REDCap electronic data capture tools hosted at CHU Clermont-Ferrand [24].

### 2.6. Statistical Methods

The sample size was determined to ensure that the confidence interval (CI) of CIPN prevalence within 5 years of chemotherapy end had an accuracy of greater than ±5%. The calculation showed that at least 400 subjects were necessary to ensure a two-sided type I error of 5%.

Statistical analysis was performed using Stata 15 (StataCorp, College Station, United States). All tests were two-sided, with a type I error set at 5%. Categorical data were presented using number of patients, percentage, and appropriate 95% CIs. Continuous data were expressed as means and standard deviations. The normality of the data was assessed using the Shapiro-Wilk test. The internal consistency of the QLQ-CIPN20 sensory scale was determined using Cronbach’s α coefficient, with a minimum accepted value of 0.70. 

Continuous data were compared between independent groups using the Student’s t-test or the Mann-Whitney U test when the assumptions of the t-test were not met. The homoscedasticity of the data was assessed using the Fisher-Snedecor test. The results were expressed using Hedge’s effect-sizes (ESs) and 95% CIs; they were interpreted according to the recommendations of Cohen [25], who defined the ES bounds as small (ES = 0.2), medium (ES = 0.5), and large (ES = 0.8). Categorical data were compared between groups using the chi-squared test or Fisher’s exact test, whereas the Stuart-Maxwell test (assessment of severity proportions of the QLQ-CIPN20 items assessing tingling, numbness, and pain in the hands and feet) was applied to compare paired proportions. To analyze the relationships between continuous parameters, Pearson and Spearman correlation coefficients were estimated according to the statistical distribution of variables and by applying Sidak’s type I error correction. As reported in the literature [26,27,28], we reported all individual *p*-values without applying systematically any mathematical correction of the aforementioned tests comparing groups. Specific attention was given to the magnitude of differences (i.e., ESs) and clinical relevance. 

To determine factors associated with the QLQ-CIPN20 sensory score (dependent variable), multivariable analyses were performed, including patient characteristics (sex, age, tobacco, alcohol, and weight variation) and characteristics of chemotherapy (chemotherapy date, cumulative dose and dose intensity of oxaliplatin, and center). More precisely, a random-effects, multiple linear model was used to take into account both between-center and within-center variability; in this, center was categorized as a random effect. The normality of residuals from these models was then analyzed. Particular attention was paid to the study of multicollinearity and interactions between covariates (1) studying the relationships between the covariables and (2) evaluating the impact to add or delete variables on multivariable model. Results are expressed as regression coefficients and 95% CIs, and forest plots were used to present the results. However, no multivariable analysis of CIPN severity and comorbidities (neuropathic pain, anxiety, depression, and HRQoL) was performed because there was a strong multicollinearity.

As less than 5% of data were missing, no handling of missing data was applied.

## 3. Results

### 3.1. Population

Four hundred and six patients were included in the study (participation rate: 68.5%) (Figure 1). The characteristics of these patients are detailed in Table 1.

### 3.2. Prevalence of Oxaliplatin-Induced Peripheral Neuropathy

The nine items of the QLQ-CIPN20 sensory scale indicated a good level of internal consistency (Cronbach’s α = 0.83). Among the recruiting centers, the mean QLQ-CIPN20 scores showed a tendency to be different (*p* = 0.06), and the cumulative doses of oxaliplatin differed significantly (*p* < 0.001).

Among all 406 patients, 31.3% (95%CI: 26.8, 36.0) had sensory CIPN. This prevalence of sensory CIPN started as 39.7% (95%CI: 28.8, 51.5) during the 1st year after chemotherapy end and decreased to 26.6% (95%CI: 16.3, 39.1) during the 5th year. The prevalence of sensory CIPN did not change over time (*p* = 0.25) (Figure 2A). The QLQ-CIPN20 sensory scores differed among the 5 years after chemotherapy end (*p* = 0.048) (Figure 2B). 

Among patients with sensory CIPN (*n* = 127), the proportions of neuropathic symptoms such as tingling, numbness, and pain were more severe in the feet than in the hands (*p* < 0.05) (Figure 3). Among all patients, 22.9% had cold hypersensitivity (VAS score: 43.7 ± 23.7) and 10.0% had heat hypersensitivity (VAS score: 42.9 ± 22.5). The proportions and VAS scores of cold hypersensitivity were higher in patients with sensory CIPN than in those without (41.7% vs. 14.2%, *p* < 0.001; 49.0 ± 23.4 vs. 36.5 ± 22.4, *p* = 0.01, ES: 0.54, 95%CI: 0.12, 0.96). The proportions, but not the VAS scores, of heat hypersensitivity were higher in patients with sensory CIPN than in those without (17.3% vs. 6.6%, *p* = 0.001 and 45.0 ± 23.7 vs. 40.2 ± 21.4, *p* = 0.5).

The QLQ-CIPN20 sensory scores were very weakly correlated with the cumulative dose and dose intensity of oxaliplatin (coefficients: 0.13 and 0.15, respectively, *p* < 0.05 in both cases), but not with the number of cycles and the mean percentage of cycles with a dose reduction. Neither the proportion of patients with sensory CIPN nor the sensory scores differed according to sex, age, or alcohol use. The sensory scores, but not the proportion of patients with CIPN, were higher among patients who smoked tobacco than among those who did not (29.9 ± 21.9 vs. 22.1 ± 20.2, *p* = 0.01, ES: 0.38, 95%CI: 0.08, 0.68). The sensory scores were not correlated with height, weight, body surface area, or BMI. However, the sensory scores were very weakly correlated with the percentage variance in weight, body surface area, and BMI between the first and last cycles (coefficients: −0.13, −0.11, and −0.13, respectively, *p* < 0.05 in all cases).

In the multivariable analysis of the QLQ-CIPN20 sensory scores and associated factors, women, tobacco smokers, and patients who lost weight during chemotherapy had higher sensory scores. Higher dose intensities of oxaliplatin were associated with higher sensory scores. Finally, older chemotherapy was associated with a lower sensory score at the time of study (Figure 4). 

Neuropathic pain was detected in 15.4% (62) of all patients and in 36.5% of patients with CIPN. The proportion of patients with neuropathic pain was higher among those with sensory CIPN than among those without CIPN (36.5% vs. 5.8%, *p* < 0.001). In the same way, patients with neuropathic pain had higher QLQ-CIPN20 sensory scores than those without (45.2 ± 19.9 vs. 19.0 ± 18.0, *p* < 0.001). The proportion of patients with neuropathic pain was not related to sex, age of patient, chemotherapy date, cumulative dose, dose intensity, or number of cycles of oxaliplatin. Among patients with sensory CIPN, 3.2% took pregabalin, 1.6% gabapentin, and 1.6% amitriptyline; none received duloxetine or imipramine (Table 2).

### 3.3. Impact of Oxaliplatin-Induced Peripheral Neuropathy

The QLQ-CIPN20 sensory scores had a weak to moderate correlation with HRQoL scores (QLQ-C30) corresponding to a degradation of HRQoL (Table 3). Patients with sensory CIPN had lower scores in global health status and functional scales, as well as higher scores in symptom scales, than those without sensory CIPN (Table 3). The highest correlations with CIPN were identified in the cases of physical functioning, role functioning, emotional functioning, fatigue, and pain. These same scales had the highest ESs among patients with and without sensory CIPN (Table 3). Notably, only pain had a large ES, while the other dimensions had medium ESs.

Patients with sensory CIPN had more symptoms of anxiety and depression than those without (Figure 5). The QLQ-CIPN20 sensory scores were higher among patients with anxiety or depression disorders (normal vs. suggestive vs. indicative scores of anxiety: 19.1 ± 18.8 vs. 25.7 ± 19.0 vs. 38.0 ± 23.2, *p* < 0.001; normal vs. suggestive vs. indicative scores of depression: 20.7 ± 19.4 vs. 26.2 ± 19.3 vs. 38.5 ± 24.4, *p* < 0.001).

## 4. Discussion

Sensory CIPN occurred in about one-third (31.3%) of patients during the 5 years after adjuvant FOLFOX-4 chemotherapy to treat colorectal cancer. The highest prevalence of sensory CIPN was identified during the 1st year (39.7%), while and the lowest (26.6%) occurred during the 5th year. The prevalence of CIPN after chemotherapy end remained high, with nearly no modification over 5 years. In the review of Seretny et al., the incidence of oxaliplatin-induced peripheral neuropathy ranged between 40.6% (95%CI: 30.8, 50.4) and 93.7% (95%CI: 81.9, 105) [29]. Soveri et al. recently reported that, among 92 patients with a median follow-up time of 4.2 years (range: 2.6–8.9), 24% still had grade 2 sensory neuropathy, while 8% had grade 3 neuropathy and 1% had grade 4 neuropathy; in total, 33% had sensory neuropathy of grade 2 or higher [30]. In another study, the prevalence of neuropathy after 25 months was of 37.5% for grade 1, 29.2% for grade 2, and 0.7% for grade 3 [16]. After 48 months, the prevalence was of 11.9% for grade 1, 2.8% for grade 2, and 0.7% for grade 3 [31]. Finally, after 8 years, 30.4% of patients had grade 2+ neuropathy [32].

Sensory CIPN is characterized by a predominance of sensory symptoms in the feet rather than the hands, as demonstrated by Yoshida et al. [33]. In the present study, 15.4% of all the patients and 36.5% of patients with sensory CIPN had neuropathic pain. De Carvalho Barbosa et al. found a pain prevalence of 20% in their cohort of oxaliplatin-treated patients; in the present cohort, the equivalent value was 15.4% [34]. It follows that neuropathic pain is probably an inefficient principal endpoint to assess antineuropathic strategies in patients with CIPN [35].

Several risk factors have been associated with sensory CIPN severity, including female sex, high dose intensity of oxaliplatin, and tobacco smoking. One study on CIPN found no difference according to sex [36]. In the case of diabetic neuropathy, females have reported a higher frequency and intensity of pain [37]. Data from the literature shows that females tend to perceive pain with more intensity and at lower pain thresholds than males do. Many of these studies found increasing trends in pain perception within the menstrual cycle, as well as trends among pre-, peri- and post-menopausal women [38]. To our knowledge, one study has demonstrated a positive link between CIPN and tobacco smoking [39], although tobacco smoking is a well-established risk factor for chronic painful conditions (for review see LaRowe and Ditre, 2020 [40]). In the case of advanced cancers, current and former smokers appeared to be significantly more likely to have higher pain levels and require higher opioid doses [41]. In previous studies, CIPN has been related to cumulative oxaliplatin dose [4,5,6], but we found that cumulative dose was not related to CIPN risk, probably because our patients had globally a higher cumulative dose of oxaliplatin (1222.3 ± 455.6 mg/m^2^) than in previous studies (695 ± 273 mg/m^2^ and 553 ± 237 mg/m^2^) [5,6], which have been reviewed [4]. Only rarely has dose intensity been associated with CIPN risk [5]. However, higher dose intensity of bortezomib or paclitaxel was related to CIPN [42,43].

As expected, the CIPN had a strong impact on anxiety, depression, fatigue, and HRQoL (physical, role, and emotional functioning) [30,44]. Sensory CIPN has been associated with depression and sleep disturbances [45]. However, Ventzel et al. found no link between CIPN and depression or anxiety assessed using the HADS [6]. In their study, the mean cumulative dose of oxaliplatin (553 ± 237 mg/m^2^) was nearly half that of the present study (1222.3 ± 455.6 mg/m^2^).

Finally, the present study underlined the difficulties in CIPN management: few of the patients were treated using analgesic drugs, and none were treated using duloxetine, which is the only recommended drug for CIPN (American Society of Clinical Oncology-ASCO) [11]. It is not clear why this management failure occurred. The ASCO guidelines for the management of CIPN [11] may not have been correctly disseminated to French oncologists. A Japanese study demonstrated that the dissemination of the Japanese Clinical Guidelines for the Management of Chemotherapy-Induced Peripheral Neuropathy in 2017 (CIPN-GL2017), incorporating ASCO recommendations, increased the prescription rate of duloxetine by Japanese oncologists, for the management of CIPN [46]. Moreover medications used for the management of peripheral neuropathic pain (antiepileptics and antidepressants) are associated to many adverse effects that may decrease patient adherence [47]. Finally, the diagnosis of CIPN is still a concern, as there is no clear consensus on a robust and easy to use tool [48], so perhaps difficulties in the diagnosis and treatment led to under-diagnosis and under-treatment of these patients. 

The use of the QLQ-CIPN20 questionnaire to assess the prevalence of CIPN is controversial. In the present study, based on a work by Alberti et al. (2014) [20], we used QLQ-CIPN20 scoring to approximate the prevalence of sensory CIPN. This self-administered questionnaire was particularly useful to assess CIPN severity using a paper questionnaire sent to patients. The sensory scores of the QLQ-CIPN20 show a high correlation with the NCI-CTCAE sensory grades (*p* < 0.001) [20]. Specifically, QLQ-CIPN20 scores between 30 and 40 (median ≈ 35; interquartile range ≈ 26–50) are associated with sensory neuropathy grade 2, while scores >40 (median ≈ 59; interquartile range ≈ 39–62) correspond to grade 3–4 [20]. Therefore, QLQ-CPIN20 scoring can discriminate neuropathy grade 1 from grade 2 (*p* < 0.001) and grade 2 from grade 3–4 (*p* < 0.001). However, it cannot differentiate grade 0 from grade 1 (*p* = 0.53) [20]. Le-Rademacher et al. (2017) concluded that there are no QLQ-CIPN20 score ranges that correspond directly with NCI-CTCAE grading levels [49]. However, they also emphasized that the QLQ-CIPN20 provided detailed information, distinguished more subtle degrees of neuropathy, and that it was more responsive to change over time than the NCI-CTCAE [49].

History of neuropathy was not recorded but would have been of interest, because it has already been described as a risk factor of CIPN [50]. Disease stage according to TNM classification and incidence of CIPN during chemotherapy were not recorded, because these data were not available in the medical prescription software. Selection bias should be limited in the present study, since the patients came from several different centers, including large and small hospitals. Information bias was probably present, since patient answers were subjective and unsupported by clinical assessment, although the oncological data came from the patient medical records and medical prescription software of each center.

## 5. Conclusions

The present real-life study showed a high prevalence of sensory CIPN in survivors of colorectal cancer. Recovery from CIPN was limited over the 5 years after the end of treatment, and the condition was associated with psychological distress. The present study also showed an unexpected failure of patient management. Oxaliplatin is a pivotal drug in the management of colorectal cancer, but its neurological safety is still a concern.

## Figures and Tables

**Figure 1 jcm-09-02400-f001:**
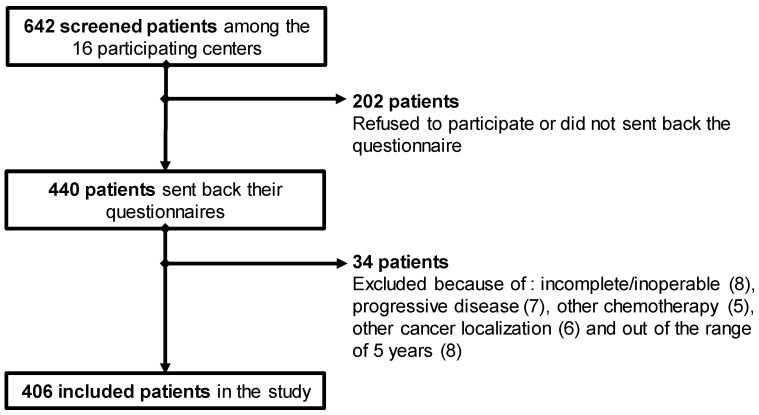
Flowchart of patient inclusion.

**Figure 2 jcm-09-02400-f002:**
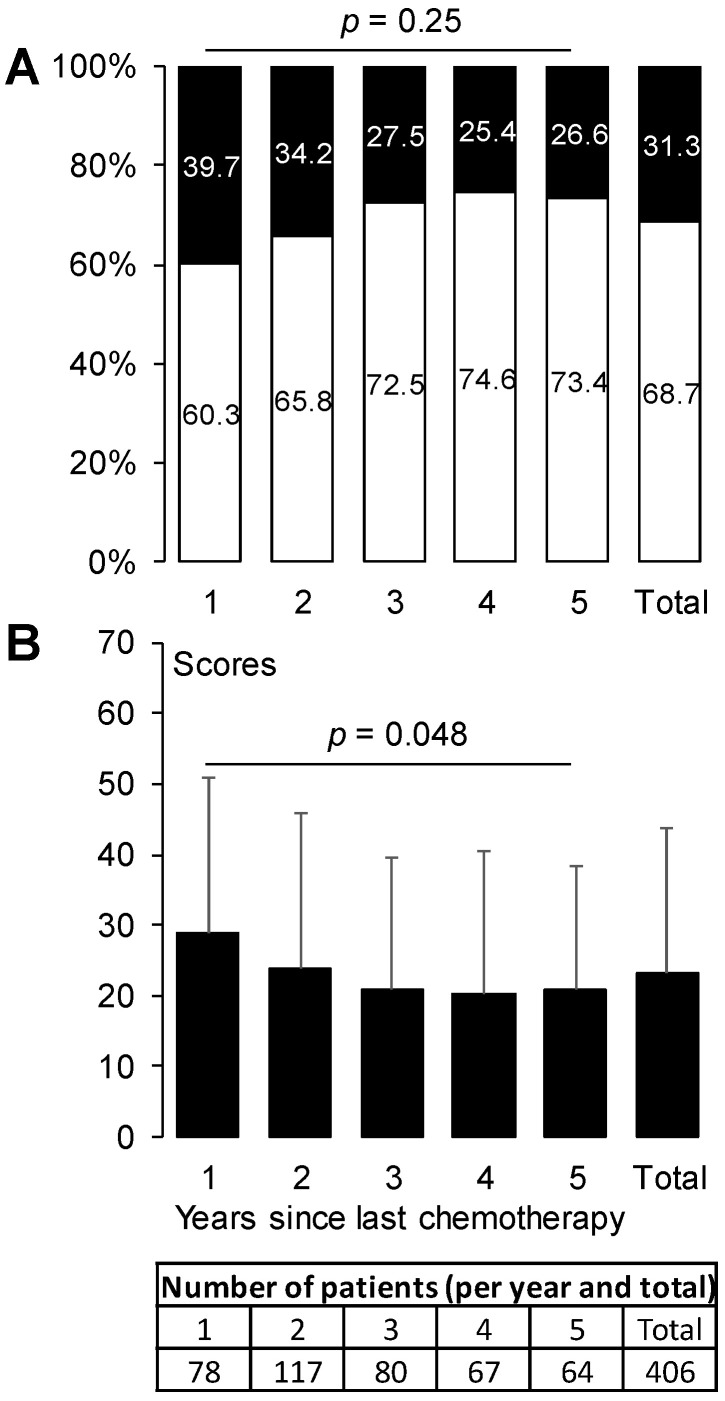
Prevalence of sensory CIPN and QLQ-CIPN20 sensory scores from 1 year until 5 years after chemotherapy end. The prevalence of sensory CIPN (**A**) is expressed as a percentage (white: no CIPN, black: sensory CIPN). The sensory scores (**B**) are expressed as mean ± standard deviation. *p*-values are provided for global comparison over time (from year 1 to year 5). CIPN, chemotherapy-induced peripheral neuropathy.

**Figure 3 jcm-09-02400-f003:**
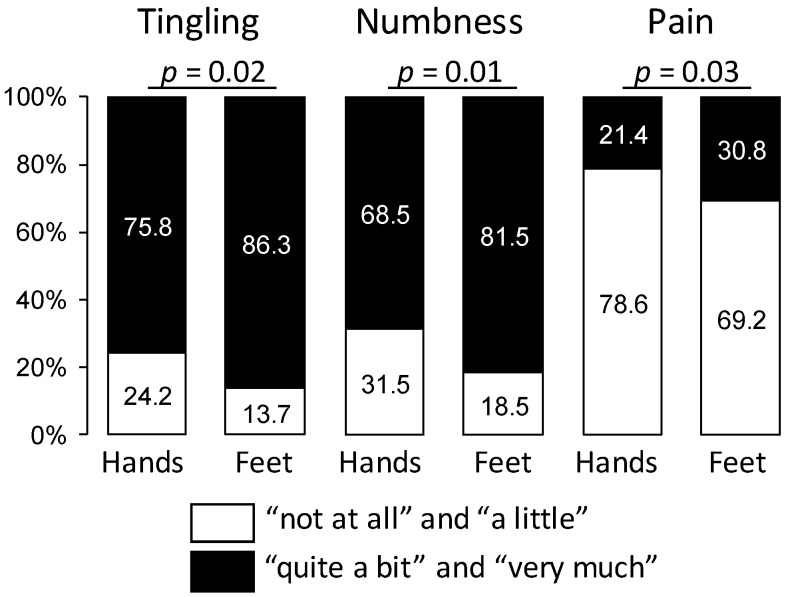
Severity proportions of the QLQ-CIPN20 items assessing tingling, numbness, and pain in the hands and feet among patients with sensory CIPN. The response categories were recoded to yield a dichotomous outcome per item (white: “not at all” or “a little” vs. black: “quite a bit” or “very much”). Results are expressed by percentage.

**Figure 4 jcm-09-02400-f004:**
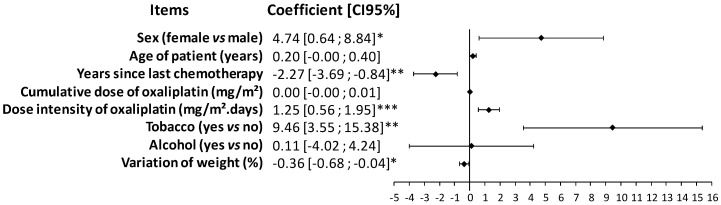
Forrest plot of the regression coefficients comparing QLQ-CIPN20 sensory scores with patient characteristics and oxaliplatin treatments. Multivariable analyses were performed, including patient characteristics (sex, age, tobacco, alcohol, and variation of weight) and characteristics of chemotherapy (chemotherapy date, cumulative dose and dose intensity of oxaliplatin, and center). * *p* < 0.05, ** *p* < 0.01, and *** *p* < 0.001.

**Figure 5 jcm-09-02400-f005:**
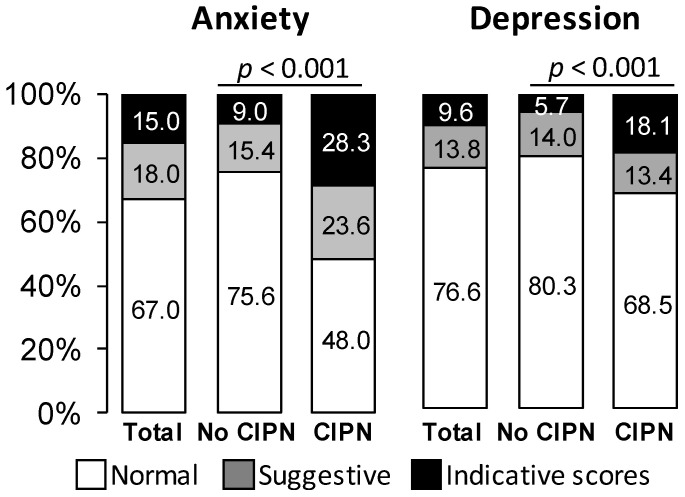
Proportion of patients with anxiety or depression (Hospital Anxiety and Depression scale) according to sensory CIPN. Results are expressed as percentages.

**Table 1 jcm-09-02400-t001:** Characteristics of the included patients (*n* = 406). Categorical variables are expressed as percentages (number). Continuous variables are expressed as mean ± standard deviation (* *p* < 0.05).

Items	All the Patients *n* = 406	No Sensory CIPN *n* = 279	Sensory CIPN *n* = 127
Male	54.2 (220)	56.6 (158)	48.8 (62)
Female	45.8 (186)	43.4 (121)	51.2 (65)
Age (year)	66.3 ± 9.7	66.0 ± 10.0	67.0 ± 8.9
	Min: 31.1; Max: 89.3	Min: 31.1; Max: 89.3	Min: 34.5; Max: 84.7
Height (cm)	168.1 ± 9.5	168.4 ± 8.9	167.4 ± 10.6
Weight (Kg)			
1st cycle	71.8 ± 16.3	72.2 ± 15.8	70.7 ± 17.4
Last cycle	72.0 ± 16.5	72.8 ± 15.9	70.4 ± 17.6
Percentage variance	0.7 ± 6.2	1.1 ± 6.3	−0.03 ± 5.9
Body mass index (kg/m^2^)			
1st cycle	25.3 ± 5.0	25.4 ± 5.0	25.1 ± 5.1
Last cycle	25.4 ± 4.9	25.6 ± 4.9	24.9 ± 4.8
Percentage variance	0.7 ± 6.2	1.1 ± 6.3	−0.03 ± 5.9
Body surface area (m^2^)			
1st cycle	1.8 ± 0.2	1.8 ± 0.2	1.8 ± 0.2
Last cycle	1.8 ± 0.2	1.8 ± 0.2	1.8 ± 0.2
Percentage variance	0.4 ± 2.7	0.5 ± 2.8	0.03 ± 2.6
Tobacco use	12.4 (50)	10.4 (29)	16.7 (21)
Alcohol use	66.3 (269)	67.4 (188)	63.8 (81)
Hazardous alcohol use (males: ≥21 units/week)	14.4 (23)	13.7 (16)	16.3 (7)
Hazardous alcohol use (females: ≥14 units/week)	12.3 (13)	11.3 (8)	14.3 (5)
Years since last chemotherapy	2.4 ± 1.6	2.5 ± 1.5	2.2 ± 1.6 *
1 year	19.2 (78)	16.9 (47)	24.4 (31)
2 years	28.8 (117)	27.6 (77)	31.5 (40)
3 years	19.7 (80)	20.8 (58)	17.3 (22)
4 years	16.5 (67)	17.9 (50)	13.4 (17)
5 years	15.8 (64)	16.9 (47)	13.4 (17)
Oxaliplatin chemotherapy			
Cumulative dose (mg/m^2)^	1220.8 ± 455.6	1201.6 ± 472.6	1263.0 ± 418.1
Number of cycles	9.7 ± 2.7	9.6 ± 2.8	9.9 ± 2.6
Dose intensity (mg/m^2^/weeks)	62.3 ± 20.9	60.9 ± 21.6	65.4 ± 18.9
Mean percentage of cycles with a dose reduction	44.8 ± 34.3	45.1 ± 35.5	44.2 ± 31.7

CIPN, chemotherapy-induced peripheral neuropathy.

**Table 2 jcm-09-02400-t002:** Type and proportion of analgesic medication use in the month prior to questionnaire completion for all patients, in patients with sensory CIPN and in those with neuropathic pain.

Drug Name	All Patients *n* = 406 % (Number)	Sensory CIPN *n* = 127 % (Number)	Neuropathic Pain *n* = 62 % (Number)
Paracetamol	26.9 (109)	29.9 (38)	37.1 (23)
Aspirin	4.2 (17)	4.7 (6)	6.5 (4)
Tramadol	3.5 (14)	4.7 (6)	6.5 (4)
Ibuprofen	3.2 (13)	3.9 (5)	4.8 (3)
Codeine + paracetamol	3.0 (12)	4.7 (6)	8.1 (5)
Tramadol + paracetamol	3.0 (12)	3.9 (5)	6.5 (4)
Opium + caffeine + paracetamol	1.7 (7)	1.6 (2)	6.5 (4)
Pregabalin	1.7 (7)	3.2 (4)	4.8 (3)
Gabapentin	1.0 (4)	1.6 (2)	4.8 (3)
Amitriptyline	1.0 (4)	1.6 (2)	3.2 (2)
Codeine	0.7 (3)	0.8 (1)	1.6 (1)
Morphine	0.7 (3)	0.8 (1)	1.6 (1)
NSAIDs other	0.7 (3)	0 (0)	1.6 (1)
Triptans	0.5 (2)	0 (0)	1.6 (1)
Opium + paracetamol	0.3 (1)	0 (0)	0 (0)
Dihydrocodeine	0 (0)	0 (0)	0 (0)
Duloxetine	0 (0)	0 (0)	0 (0)
Imipramine	0 (0)	0 (0)	0 (0)
Oxycodone	0 (0)	0 (0)	0 (0)

**Table 3 jcm-09-02400-t003:** Comparison of QLQ-C30 score according to sensory CIPN and correlation between QLQ-C30 score and QLQ-CIPN20 sensory score. The correlations between the QLQ-C30 score and QLQ-CIPN20 sensory score are presented with the correlation coefficient and significance.

Dimensions	QLQ-C30 Scores	QLQ-C30 Scores According to Sensory CIPN			Correlations QLQ-C30 and Sensory Scores
		No Sensory CIPN	Sensory CIPN	Effect Size (95% CI)	
Global health status	69.2 ± 20.5	71.6 ± 20.1	63.7 ± 20.4 ***	−0.39 (−0.61, −0.18)	−0.28 ^#^
Physical functioning	81.7 ± 19.7	84.7 ± 17.8	75.0 ± 22.1 ***	−0.50 (−0.71, −0.29)	−0.32 ^#^
Role functioning	79.1 ± 28.2	84.2 ± 24.5	67.9 ± 32.3 ***	−0.60 (−0.81, −0.38)	−0.38 ^#^
Emotional functioning	74.6 ± 25.1	78.8 ± 22.5	65.2 ± 27.8 ***	−0.56 (−0.77, −0.34)	−0.32 ^#^
Cognitive functioning	77.6 ± 23.5	80.7 ± 21.0	70.7 ± 27.2 ***	−0.43 (−0.64, −0.22)	−0.22 ^#^
Social functioning	74.3 ± 30.5	78.6 ± 28.2	64.5 ± 33.1 ***	−0.47 (−0.69, −0.26)	−0.28 ^#^
Fatigue	33.7 ± 27.2	28.3 ± 24.6	45.5 ± 29.0 ***	0.66 (0.44, 0.87)	0.36 ^#^
Nausea and vomiting	6.5 ± 15.7	5.7 ± 14.8	8.4 ± 17.4	0.17 (0.04, 0.38)	0.17 ^#^
Pain	20.5 ± 26.0	14.0 ± 22.2	34.6 ± 28.1 ***	0.85 (0.63, 1.07)	0.42 ^#^
Dyspnea	23.2 ± 28.1	21.0 ± 26.6	28.0 ± 30.8 *	0.25 (0.04, 0.46)	0.18 ^#^
Insomnia	34.8 ± 35.4	29.1 ± 33.4	47.4 ± 36.4 ***	0.53 (0.32, 0.74)	0.28 ^#^
Appetite loss	11.2 ± 23.6	8.9 ± 20.6	16.4 ± 28.5 **	0.32 (0.11, 0.53)	0.21 ^#^
Constipation	19.3 ± 28.2	17.1 ± 26.8	24.1 ± 30.9 *	0.25 (0.04, 0.46)	0.14 ^#^
Diarrhea	24.2 ± 31.9	20.7 ± 29.5	31.7 ± 35.5 **	0.35 (0.14, 0.56)	0.23 ^#^
Financial difficulties	9.9 ± 22.2	6.9 ± 17.0	16.5 ± 29.8 **	0.44 (0.23, 0.65)	0.20 ^#^

* *p* < 0.05, ** *p* < 0.01, and *** *p* < 0.001 for no sensory CIPN vs. sensory CIPN; # *p* < 0.05.

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
