# Peer review of "Long-Term Prevalence of Sensory Chemotherapy-Induced Peripheral Neuropathy for 5 Years after Adjuvant FOLFOX Chemotherapy to Treat Colorectal Cancer: A Multicenter Cross-Sectional Study"

_jcm, 2020, doi:10.3390/jcm9082400_

Round 1
Reviewer 1 Report
Thank you for your very interesting manuscript.
- Participants: "The exclusion criteria were age < 18 years and neurological diseases (stroke, Parkinson’s
131 disease, Alzheimer’s disease)" would be "The exclusion criteria were age > 18 years and neurological diseases (diabetic neuropathy, stroke, Parkinson’s disease, Alzheimer’s disease)". - You utilized the QLQ142 CIPN20 for the evaluation of oxaliplatin-induced peripheral neuropathy. The cut-off score of ≥ 30/100 could be explained a little bit more in detail.
Author Response
Q1. Participants: "The exclusion criteria were age < 18 years and neurological diseases (stroke, Parkinson’s disease, Alzheimer’s disease)" would be "The exclusion criteria were age > 18 years and neurological diseases (diabetic neuropathy, stroke, Parkinson’s disease, Alzheimer’s disease)".
A1. Probably that a comma is missing. We were not able to include young patients (age below 18 years). So the following modifications has been done (lines 131-132): “The exclusion criteria were age < 18 years, and patients with neurological diseases (stroke, Parkinson’s disease, Alzheimer’s disease).”
Q2. You utilized the QLQ142 CIPN20 for the evaluation of oxaliplatin-induced peripheral neuropathy. The cut-off score of ≥ 30/100 could be explained a little bit more in detail.
A2. The choice of this cut-off is now detailed in the discussion part (as request by a reviewer lines 363-373). According to Alberti et al. (2013), this cut-off score of ≥ 30/100 correspond to NCI-CTCAE sensory grade of 2.
“The sensory scores of the QLQ-CIPN20 show a high correlation with the NCI-CTCAE sensory grades (p < 0.001) [20]. Specifically, QLQ-CIPN20 scores between 30 and 40 (median ≈ 35; interquartile range ≈ [26;50]) are associated with sensory neuropathy grade 2, while scores > 40 (median ≈ 59; interquartile range ≈ [39;62]) correspond to grade 3-4 [20]. Therefore, QLQ-CPIN20 scoring can discriminate neuropathy grade 1 from grade 2 (p < 0.001), and grade 2 from grade 3-4 (p < 0.001). However, it cannot differentiate grade 0 from grade 1 (p = 0.53) [20]. Le-Rademacher et al. (2017) concluded that there are no QLQ-CIPN20 score ranges that correspond directly with NCI-CTCAE grading levels [49]. However, they also emphasized that the QLQ-CIPN20 provided detailed information, distinguished more subtle degrees of neuropathy, and that it was more responsive to change over time than the NCI-CTCAE [49].”
Reviewer 2 Report
Review of the manuscript entitled “Long-term prevalence of sensory chemotherapy- induced peripheral neuropathy for 5 years after adjuvant FOLFOX chemotherapy to treat colorectal cancer: A multicenter cross-sectional study”:
In this study, the authors assessed the prevalence and severity of CIPN after oxaliplatin chemotherapy in patients with colorectal cancer in a multicentre study with 16 participating centres. Additionally they assessed neuropathic pain, anxiety, depression, health related quality of life and the use of pain medications.
The paper is written in a precise and straightforward manner. The introduction gives all necessary background information and Leeds to the hypothesis which is clearly formulated. The methods are precisely described and adequate to address the hypothesis. The results are displayed in an well-arranged manner. The discussion comprises the relevant literature.The conclusions are adequate and supported by the data.
I have only minor proposals for improvement:
L 141 -151 Consider placing this paragraph in the discussion of the methods.
L153 DN4 should be explained.
L194 I am not quite sure, whether it is adequate, not to perform multiple adjustment corrections, even if there are references indicating that this is the case. Maybe it is helpful to quote the paper by Bender and Lange (J Clin Epid, 2001), as they do not generally object the use of multiple adjustment corrections but define situations when to use and when not to use them.
L345 ff
The most astonishing finding of the study is how scarcely specific medications are used in CIPN. Are they further possible explanations. Maybe consider inserting a sentence proposing on how to solve this issue.
L347 insert “CIPN” after “with”
Author Response
Q3. L 141 -151 Consider placing this paragraph in the discussion of the methods.
A3. This paragraph has been added to the discussion part (lines 360-373): “The use of the QLQ-CIPN20 questionnaire to assess the prevalence of CIPN is controversial. In the present study, based on a work by Alberti et al. (2014) [20], we used QLQ-CIPN20 scoring to approximate the prevalence of sensory CIPN. This self-administered questionnaire was particularly useful to assess CIPN severity using a paper questionnaire sent to patients. The sensory scores of the QLQ-CIPN20 show a high correlation with the NCI-CTCAE sensory grades (p < 0.001) [20]. Specifically, QLQ-CIPN20 scores between 30 and 40 (median ≈ 35; interquartile range ≈ [26;50]) are associated with sensory neuropathy grade 2, while scores > 40 (median ≈ 59; interquartile range ≈ [39;62]) correspond to grade 3-4 [20]. Therefore, QLQ-CPIN20 scoring can discriminate neuropathy grade 1 from grade 2 (p < 0.001), and grade 2 from grade 3-4 (p < 0.001). However, it cannot differentiate grade 0 from grade 1 (p = 0.53) [20]. Le-Rademacher et al. (2017) concluded that there are no QLQ-CIPN20 score ranges that correspond directly with NCI-CTCAE grading levels [49]. However, they also emphasized that the QLQ-CIPN20 provided detailed information, distinguished more subtle degrees of neuropathy, and that it was more responsive to change over time than the NCI-CTCAE [49].”
Q4. L153 DN4 should be explained.
A4. The requested explanation has been added (lines 145-147): “With regards to the secondary endpoints, ongoing neuropathic pain was defined as a visual analogue scale (VAS) score ≥ 40/100 and a DN4 (French abbreviation: Douleur Neuropathique 4, for neuropathic pain 4) interview questionnaire score ≥ 3/7 [21].”
Q5. L194 I am not quite sure, whether it is adequate, not to perform multiple adjustment corrections, even if there are references indicating that this is the case. Maybe it is helpful to quote the paper by Bender and Lange (J Clin Epid, 2001), as they do not generally object the use of multiple adjustment corrections but define situations when to use and when not to use them.
A5. We thank the reviewer for the interesting comment and relevant reference which confirms our choice. On the other hand, in exploratory studies, in which data are collected with an objective but not with a prespecified key hypothesis, multiple test adjustments are not strictly required. Other investigators hold an opposite position that multiplicity corrections should be performed in exploratory studies. We agree that the multiplicity problem in exploratory studies is huge. However, the use of multiple test procedures does not solve the problem of making valid statistical inference for hypotheses that were generated by the data. Exploratory studies frequently require a flexible approach for design and analysis. The choice and the number of tested hypotheses may be data dependent, which means that multiple significance tests can be used only for descriptive purposes but not for decision making, regardless of whether multiplicity corrections are performed or not. As the number of tests in such studies is frequently large and usually a clear structure in the multiple tests is missing, an appropriate multiple test adjustment is difficult or even impossible. Hence, we prefer that data of exploratory studies be analyzed without multiplicity adjustment. According to the reviewer’s comment, we added Bender and Lange’s reference in Statistics section (lines 188-189): “As reported in the literature [26–28], we reported all individual p-values without applying systematically any mathematical correction of the aforementioned tests comparing groups.”
Q6. L345 ff
A6. Sorry but we don’t understand what the reviewer means by “ff”
Q7. The most astonishing finding of the study is how scarcely specific medications are used in CIPN. Are they further possible explanations. Maybe consider inserting a sentence proposing on how to solve this issue.
A7. Yes it’s right, we have no clear explanation for this result, but only hypothesis. From discussion with patients and oncologists, pain medications for CIPN (neuropathic pain) are considered not effective with many adverse effects. Patients can say that they prefer to have tingling instead of dizziness. Another hypothesis could be that these patients are mainly managed par general practitioner who are probably not informed of recommended strategies for CIPN (duloxetine). In France, we know that oncologists mainly use pregabalin (we have performed a study on French oncologist’s practices, and the results are under review in a scientific journal). Moreover, the diagnosis of CIPN is still a concern in clinical practice, as in clinical research, because no consensus has been clearly defined for CIPN.
We have modified the discussion accordingly (lines 347-373): “Finally, the present study underlined the difficulties in CIPN management: few of the patients were treated using analgesic drugs, and none were treated using duloxetine, which is the only recommended drug for CIPN (American Society of Clinical Oncology-ASCO) [11]. It is not clear why this management failure occurred. The ASCO guidelines for the management of CIPN [11] may not have been correctly disseminated to French oncologists. A Japanese study demonstrated that the dissemination of the Japanese Clinical Guidelines for the Management of Chemotherapy-Induced Peripheral Neuropathy in 2017 (CIPN-GL2017), incorporating ASCO recommendations, increased the prescription rate of duloxetine by Japanese oncologists, for the management of CIPN [46]. Moreover medications used for the management of peripheral neuropathic pain (antiepileptics and antidepressants) are associated to many adverse effects that may decrease patient adherence [47]. Finally, the diagnosis of CIPN is still a concern, as there is no clear consensus on a robust and easy to use tool [48], so perhaps difficulties in the diagnosis and treatment led to under-diagnosis and under-treatment of these patients.”
Q8. L347 insert “CIPN” after “with”
A8. This paragraph has been modified for more details (lines: 347-373): “Finally, the present study underlined the difficulties in CIPN management: few of the patients were treated using analgesic drugs, and none were treated using duloxetine, which is the only recommended drug for CIPN (American Society of Clinical Oncology-ASCO) [11]. It is not clear why this management failure occurred. The ASCO guidelines for the management of CIPN [11] may not have been correctly disseminated to French oncologists. A Japanese study demonstrated that the dissemination of the Japanese Clinical Guidelines for the Management of Chemotherapy-Induced Peripheral Neuropathy in 2017 (CIPN-GL2017), incorporating ASCO recommendations, increased the prescription rate of duloxetine by Japanese oncologists, for the management of CIPN [46]. Moreover medications used for the management of peripheral neuropathic pain (antiepileptics and antidepressants) are associated to many adverse effects that may decrease patient adherence [47]. Finally, the diagnosis of CIPN is still a concern, as there is no clear consensus on a robust and easy to use tool [48], so perhaps difficulties in the diagnosis and treatment led to under-diagnosis and under-treatment of these patients.”
Reviewer 3 Report
In this study, Marie Selvy et al. observed the long-term prevalence of sensory chemotherapy-induced peripheral neuropathy for 5 years after adjuvant FOLFOX chemotherapy. This manuscript is interesting; however, in my opinion, the description of the included patients and the discussion part seems to be weak to explain/support the results. Below are my comments.
1) In table 1, the female population should also be mentioned.
2) Can authors please specify 'alcohol units/week'?
3) This manuscript focused on the long-term prevalence of peripheral neuropathy. However, the information related to oxaliplatin treatment (i.e. cumulative doses, number of cycles) does not match to the patients divided to 'Years since last chemotherapy'. It will be more informative if authors can precise the average treatment that patients have received in each group.
4) As mentioned in the discussion, the disease stage according to TNM classification was not mentioned in this study. But, does the disease stage may not affect the degree of the peripheral neuropahty? There are any references concerning this point?
5) The results are interesting. However, the explanation that further support or explain these results seems to be poor. Can authors please give deepened putative reasons why sex, and high dose intensity, and especially tobacco smoking are associated with CIPN?
6) Does the drugs that patients have taken have no effect on CIPN and on the results? It will be interesting to know how long (also the doses used) the patients have taken their pills.
Author Response
Q9. 1) In table 1, the female population should also be mentioned.
A9. The requested information has been added to the table 1:
“Male: 54.2 (220)”
“Female: 45.8 (186)”
Q10. 2) Can authors please specify 'alcohol units/week'?
A10. The requested information has been added to the table 1
“Hazardous alcohol use (males: ≥ 21 alcohol units/week): 14.4 (23)”
“Hazardous alcohol use (females: ≥ 14 alcohol units/week): 12.3 (13)”
Q11. 3) This manuscript focused on the long-term prevalence of peripheral neuropathy. However, the information related to oxaliplatin treatment (i.e. cumulative doses, number of cycles) does not match to the patients divided to 'Years since last chemotherapy'. It will be more informative if authors can precise the average treatment that patients have received in each group.
A11. We add to the table 1 all the details on patients’ characteristics and treatments for the patients without sensory CIPN and the patients with a sensory CIPN.
Q12. 4) As mentioned in the discussion, the disease stage according to TNM classification was not mentioned in this study. But, does the disease stage may not affect the degree of the peripheral neuropathy? There are any references concerning this point?
A12. The TNM was not available in the medical prescription software, and was consequently not recorded for this study. However, all the included patients received an adjuvant FOLFOX chemotherapy (cancer stage II and III) and were in remission during the survey (no cancer relapse). We did not find any publication that compared prevalence or severity of CIPN in non-metastatic versus metastatic patients. However this comment is very relevant since we can expect a higher pain level in metastatic patients and also a higher prevalence of CIPN because of higher cumulative doses of oxaliplatin.
Q13. 5) The results are interesting. However, the explanation that further support or explain these results seems to be poor. Can authors please give deepened putative reasons why sex, and high dose intensity, and especially tobacco smoking are associated with CIPN?
A13. Literature on sex difference for CIPN is scarce and in most of cases without significant difference. This lack of difference is probably related to the small sample size and also due to the fact that statistical analysis were univariate. Our multivariable analysis should have neutralized confounding factors between males and females, and show a difference in CIPN severity. Data from the literature show that females tend to perceive pain with more intensity and at lower pain thresholds than males do. In addition, many of these studies found increasing trends in pain perception within the menstrual cycle, as well as trends among pre-, peri- and post-menopausal women (Maurer et al., 2016).
Lines 328-332: “In the case of diabetic neuropathy, females have reported a higher frequency and intensity of pain [37]. Data from the literature shows that females tend to perceive pain with more intensity and at lower pain thresholds than males do. Many of these studies found increasing trends in pain perception within the menstrual cycle, as well as trends among pre-, peri- and post-menopausal women [38].”
To our knowledge, there is only one publication on tobacco and CIPN (Kawakami et al., 2012). However, tobacco is a very well established risk factor of painful condition (for review see LaRowe and Ditre, 2020).
Lines 332-336: “To our knowledge, one study has demonstrated a positive link between CIPN and tobacco smoking [39], although tobacco smoking is a well-established risk factor for chronic painful conditions (for review see LaRowe and Ditre, 2020 [40]). In the case of advanced cancers, current and former smokers appeared to be significantly more likely to have higher pain levels and require higher opioid doses [41].”
Few studies have assessed the relation between oxaliplatin dose intensity and CIPN, and found no relation. But these studies had small sample sizes (Tanishima et al., 2017; Beijers et al., 2015). However, higher dose intensity of bortezomib or paclitaxel was related to CIPN (Li et al., 2019 and Hertz et al., 2018).
Lines 340-341: “However, higher dose intensity of bortezomib or paclitaxel was related to CIPN [42,43].”
Q14. 6) Does the drugs that patients have taken have no effect on CIPN and on the results? It will be interesting to know how long (also the doses used) the patients have taken their pills.
A14. We do not know the doses and durations of analgesic treatments. It was only asked to patients if they were taking analgesics drugs during the past month. Moreover, the number of patients taking pain medication is too limited to perform statistical analysis (1-3% of patients taking medication for neuropathic pain, e.g. pregabalin, gabapentin, amitriptyline).
Round 2
Reviewer 3 Report
The authors have answered to all questions in detail. Thus, I suggest that this manuscript is published in the present form.